# Growth Hormone Alters Circulating Levels of Glycine and Hydroxyproline in Mice

**DOI:** 10.3390/metabo13020191

**Published:** 2023-01-28

**Authors:** Jonathan A. Young, Silvana Duran-Ortiz, Stephen Bell, Kevin Funk, Yuan Tian, Qing Liu, Andrew D. Patterson, Edward O. List, Darlene E. Berryman, John J. Kopchick

**Affiliations:** 1Heritage College of Osteopathic Medicine, Ohio University, Athens, OH 45701, USA; 2Edison Biotechnology Institute, Ohio University, Athens, OH 45701, USA; 3Department of Veterinary and Biomedical Sciences, The Pennsylvania State University, University Park, State College, PA 16803, USA

**Keywords:** growth hormone, metabolomics, amino acids, transcriptomics

## Abstract

Growth hormone (GH) has established effects on protein metabolism, such as increasing protein synthesis and decreasing amino acid degradation, but its effects on circulating amino acid levels are less studied. To investigate this relationship, metabolomic analyses were used to measure amino acid concentrations in plasma and feces of mice with alterations to the GH axis, namely bovine GH transgenic (bGH; increased GH action) and GH receptor knockout (GHRKO; GH resistant) mice. To determine the effects of acute GH treatment, GH-injected GH knockout (GHKO) mice were used to measure serum glycine. Furthermore, liver gene expression of glycine metabolism genes was assessed in bGH, GHRKO, and GH-injected GHKO mice. bGH mice had significantly decreased plasma glycine and increased hydroxyproline in both sexes, while GHRKO mice had increased plasma glycine in both sexes and decreased hydroxyproline in males. Glycine synthesis gene expression was decreased in bGH mice (Shmt1 in females and Shmt2 in males) and increased in GHRKO mice (Shmt2 in males). Acute GH treatment of GHKO mice caused decreased liver Shmt1 and Shmt2 expression and decreased serum glycine. In conclusion, GH alters circulating glycine and hydroxyproline levels in opposing directions, with the glycine changes at least partially driven by decreased glycine synthesis.

## 1. Introduction

Growth hormone (GH) is a peptide hormone that exerts effects throughout the body directly or indirectly through its major mediator, insulin-like growth factor 1 (IGF-1). In addition to the growth promotion for which it is named, GH has numerous metabolic effects, including influencing macronutrient and energy metabolism. For example, GH has well-established diabetogenic properties (one of the first described effects of GH) [1], influencing carbohydrate metabolism, and has well-established lipolytic and anti-lipogenic effects, especially in adipose tissue. GH’s role in protein metabolism is also well documented [2]. That is, one of the primary anabolic effects of GH is increased protein synthesis, especially in the muscle, with decreased amino acid degradation and decreased urea formation [2]. Further, GH preserves protein levels during fasting by stimulating lipolysis, allowing fat catabolism to serve as a primary energy source [3].

The interplay between GH and amino acids is illustrated by the ability of particular amino acids—lysine, arginine, and ornithine—to simulate GH release [4]. However, GH also influences the metabolic fate of specific amino acids, which has not been studied as well. While acute GH treatment in healthy adults does not generally cause changes in serum amino acid concentrations [3], results are not consistent, with one study showing decreased phenylalanine concentration post-GH treatment [5]. Moreover, shifts in specific amino acids have been reported in the serum of individuals with extremes in GH action [6,7]. For example, patients with GH deficiency exhibit a slight decrease in cysteine and marked increases in glutamic acid, which normalize with disease management [6]. On the other end of the spectrum, patients with increased serum GH and IGF-1 levels (acromegaly) have fewer circulating branched-chain amino acids (namely valine and isoleucine) and lysine compared to controls [7].

Mouse models have been developed to examine the effects of chronic GH alterations. Bovine GH transgenic (bGH) mice overexpress GH and possess increased levels of IGF-1, resulting in increased body mass and length and lean body composition [8], similar to acromegaly. Despite their leanness, bGH mice are insulin resistant and have a dramatically shortened lifespan [9]. In contrast, GH receptor knockout (GHRKO) mice are GH resistant and exhibit very low levels of IGF-1 and high levels of GH, causing decreased body mass, length, and obesity [8,10]. In contrast to the bGH mice, GHRKO mice are extremely insulin sensitive and have increased lifespan [11], including the designation of the longest-lived laboratory mouse [12]. These mice represent two extremes in GH action, so they are useful in identifying in vivo GH’s direct or indirect effects on metabolism and aging.

Besides disruption of the GH axis, lifespan extension in mice has also been accomplished through the modulation of dietary amino acids. The use of diets with increased glycine [13] or decreased methionine [14] has been associated with increased lifespan, with a larger lifespan increase in the latter diet. Since the effects of GH on amino acid metabolism are not fully understood and since GH and amino acid metabolism are both linked to longevity, the purpose of this study was to determine whether GH action altered circulating amino acid levels in bGH and GHRKO mice.

## 2. Materials and Methods

Mouse husbandry—Bovine GH (bGH) transgenic and GH receptor knockout (GHRKO) mice were bred or backcrossed at least 10 times onto a C57Bl/6J background, as previously described [8,11]. Mice aged 7 months were used, with *n* = 6 or 7. Mice were bred and housed at Edison Biotechnology Institute at Ohio University, kept on a 14 h light/10 h dark cycle, housed up to four mice per cage, and given ad libitum access to water and ProLab RMH 3000 chow. All mouse procedures were approved by the Ohio University Institutional Animal Care and Use Committee.

Blood collection and plasma processing—After a 6 h fast, blood was collected in heparinized tubes from the tail vein before being transferred to a microcentrifuge tube and centrifuged at 8000× *g* for 10 min at 4 °C. Plasma was collected from the supernatant and was stored at −80 °C and shipped to Penn State University for metabolomics testing.

AccQ•Tag Amino Acid Analysis of Plasma Samples—Amino acids were extracted from 20 μL plasma samples with 500 μL of ice-cold methanol/water (2:1) solution (containing 5 μM of Norvaline), followed by vortex and centrifugation (Eppendorf, Hamburg, Germany), the supernatant was collected, evaporated to dryness (Thermo Scientific, Waltham, MA, USA) and then resuspended in 50 μL 0.1N HCl solution. Amino acid derivation with AccQ•Tag reagents (Waters, Milford, MA, USA) was conducted according to the manufacturer’s protocol. Briefly, 10 μL of plasma extract was mixed with 70 μL of AccQ•Tag Ultra borate buffer and 20 μL of AccQ•Tag Ultra reagent in Total Recovery Vial. The vials were capped and vortex for several seconds and proceeded for 10 min at 55 °C. Amino Acids were detected by Waters Xevo TQS coupled with PDA, and an AccQTag Ultra Column (C18 1.7 μm 2.1 × 100 mm) with in-line filter (Waters, Milford, MA, USA) was used for separation. The results were quantified by comparing integrated peak areas against a standard curve.

NMR Analysis of Plasma Samples—100 μL plasma samples were extracted with 300 μL of ice-cold methanol; after evaporation via a SpeedVac vacuum concentrator (Thermo Scientific, Waltham, MA, USA), the extracts were resuspended in 600 μL PBS [0.1 M, 50% (*v*/*v*) D2O, and 0.005% sodium 3-trimethylsilyl [2,2,3,3-d4] propionate (TSP)]. 1H NMR spectra were acquired on a Bruker Avance NEO 600 MHz spectrometer equipped with an inverse cryogenic probe (Bruker Biospin, Billerica, MA, USA) at 298 K. A typical 1D NMR spectrum named NOESYPR1D was acquired for each sample. The metabolites were assigned on the basis of published results [15] and confirmed with a series of 2D NMR spectra. All 1H NMR spectra were adjusted for phase and baseline using Chenomx (Chenomx Inc, Edmonton, AB, Canada). The chemical shift of 1H NMR spectra was referenced to TSP at δ 0.00. The quantification of metabolites in plasma was calculated by NMR peak area against TSP using Chenomx.

Fecal Sample Collection—bGH and GHRKO mice were fasted for 6 h, and fecal samples were collected as described previously [16,17]. Briefly, mice were allowed to defecate naturally on a sterilized surface or encouraged to defecate through gentle massage in the hind-back region. Fecal pellets were collected with sterilized forceps and immediately frozen on dry ice, and held at −80 °C until shipment.

Fecal Sample Analysis- Fecal samples were analyzed as described previously [18].

qPCR—Mice were anesthetized with CO_2_ and sacrificed using cervical dislocation. Liver tissue was dissected, flash-frozen in liquid nitrogen, and stored at −80 °C until RNA isolation. RNA was isolated using a GeneJet RNA isolation kit according to manufacturer’s instructions. cDNA was synthesized using an Applied Biosystems High-Capacity cDNA Reverse Transcription kit, and quantitative PCR was performed using Power SYBR Green Mastermix on a QuantStudio 3 Thermal Cycler. Primer sequences are listed in supplemental data. qPCR data analysis was performed using qBase+ version 2.3.

RNASeq analysis of GH-Injected Mice—Male GH knockout mice generated as previously described [19] were injected at 4 months of age with recombinant human GH (5 μg/g body weight, Protein Laboratories Rehovot) or vehicle daily for five days and dissected four hours after the final injection. RNA was extracted from frozen liver tissue using a QIAGEN RNeasy mini kit according to manufacturer’s instructions. RNA sequencing was performed as described previously [19].

Fluorometric Measurement of Serum Glycine—Blood was collected from hGH or vehicle-injected GHKO mice by retroorbital bleeding at the time of dissection and was held at room temperature to allow it to clot before being centrifuged at 8000× *g* for 10 min at 4 °C. Serum was collected from the supernatant and was stored at −80 °C until use. Glycine levels in the serum were measured using the Cell Biolabs, Inc. (San Diego, CA, USA) Glycine Assay Kit (Cat# Met-5070) according to manufacturer’s instructions.

Statistics—For qPCR analysis, CNRQ values were obtained from qBase+ and tested for normality using the Shapiro–Wilks test. For comparisons between experimental and control groups, a *t*-test was used for normal samples, and a Mann–Whitney U test was used when one or both groups failed the normality test. Data are reported as mean ± SEM.

## 3. Results

### 3.1. GH Action Is Positively Associated with Plasma Hydroxyproline and Negatively Associated with Plasma Glycine Levels

To assess the relationship between GH action and circulating levels of amino acids, plasma was collected from two mouse lines with altered GH action. bGH transgenic (GH overexpression) and GHRKO (GH resistance) mice were used to represent two extremes of GH action, and both males and females were included. Twenty-three amino acids were measured, with only glycine significantly changed in both sexes and in both genotypes compared to wild-type controls (Figure 1). Notably, glycine and hydroxyproline were changed in opposite directions at each end of the spectrum of GH action, with hydroxyproline increased in bGH mice (Males 1.82 fold, *p* = 1 × 10^−6^; Females 2.08 fold, *p* = 1 × 10^−5^) and decreased in GHRKO male mice (1.49 fold, *p* = 2 × 10^−4^). In contrast, glycine was decreased in bGH mice (Males 2.06 fold, *p* = 0.001; Females 2.31 fold, *p* = 0.002) and increased in GHRKO mice (Males 1.70 fold, *p* = 0.003; Females 1.93 fold, *p* = 0.034).

Other amino acids were significantly altered in one or more groups, with bGH mice of both sexes having increased arginine, asparagine, lysine, ornithine, and threonine, as well as decreased tyrosine. Some changes were limited to a single group, with bGH males having increased alanine and tryptophan and decreased taurine and bGH females having increases in aspartate, glutamate, methionine, and proline. GHRKO males had increased tryptophan.

### 3.2. Chronic GH Alterations Have No Effect on Glycine Synthesis or Disposal

To determine the cause of the altered plasma glycine in mice with altered GH action, the liver expression of glycine synthesis (SHMT1 and SHMT2) and catabolism (GLDC and GSS) genes was assessed (Figure 2A–D). There were no genes that were consistently altered across genotypes and sexes. SHMT1 was significantly decreased in female bGH mice (*p* = 0.03), while SHMT2 was decreased in male bGH mice (*p* = 0.009) and increased in male GHRKO mice (*p* = 0.0004). GLDC expression was decreased in bGH mice (*p* = 0.005), while there were no significant changes in GSS gene expression. Glycine disposal was evaluated through fecal amino acid measurement, with no significant changes to fecal glycine levels within each sex (Figure 2E).

### 3.3. Acute GH Treatment Decreases Expression of Glycine Synthesis Genes

Because bGH and GHRKO mice have life-long (chronic) GH alterations, it is difficult to determine whether the glycine changes in these mice are directly linked to GH or just correlated. To address this, the response of glycine metabolism to an acute GH treatment was assessed by injecting recombinant human GH into mice that do not express pituitary GH (GHKO). Liver expression of two major glycine synthesis genes, SHMT1 and SHMT2, was significantly decreased (q = 9.08 × 10^−14^ and q = 1.34 × 10^−7^, respectively) following five daily injections of human GH (Figure 3A). When serum levels of glycine were assessed in these mice, there was also a significant decrease in serum glycine in GH-treated mice compared to controls after only five days of GH treatment.

## 4. Discussion

GH increases protein synthesis in tissues, but its effect on circulating amino acid levels is more complex and less well-studied. When GH action is increased, as in bGH transgenic mice, plasma glycine levels are decreased, and hydroxyproline levels are increased. In GHRKO mice, which are GH-resistant, the opposite is true, with increased plasma glycine and decreased plasma hydroxyproline (in males). In these models of chronic GH alterations, glycine synthesis (Shmt1 and Shmt2) gene expression changes correspond to plasma glycine, but many of the changes are not statistically significant. However, upon acute GH administration, GH-deficient mice have a significant decrease in the expression of glycine synthesis genes in the liver resulting in decreased serum glycine. Thus, GH action and circulating glycine levels are linked, possibly through alterations of glycine synthesis gene expression.

Although glycine and hydroxyproline are the only amino acids significantly altered in at least three of the four groups, other amino acids are significantly changed in both sexes of one of the models. In bGH mice, arginine, asparagine, lysine, ornithine, and threonine are increased, and tyrosine is decreased in the plasma of both sexes. Some of these results contrast with previous reports, which have shown decreases in circulating lysine [20] and arginine [21] upon GH treatment. Interestingly, arginine, lysine, and ornithine are each well-established drivers of GH secretion [4], so the increase of these amino acids in circulation in bGH mice indicates that there may be a bidirectional link between GH secretion and plasma levels of these three amino acids.

Serine is not significantly changed in any groups. This is notable because serine is a major nondietary source of glycine by serving as a substrate for Shmt2, a gene whose product is the major source of glycine synthesis in the liver [22]. In these models, plasma serine does not seem to change concomitantly with plasma glycine.

Plasma hydroxyproline levels in this study are positively associated with GH action; bGH mice have increased circulating hydroxyproline, and GHRKO mice have decreased plasma hydroxyproline. This finding was expected, as the association between GH and fibrosis is well established [23], and hydroxyproline is an amino acid that is specific to collagen, which is increased in fibrosis. Urinary hydroxyproline excretion was previously used as a marker for bone resorption [24], and humans with acromegaly (excess GH) have increased bone turnover [25], which is another possible explanation for the increased plasma hydroxyproline. However, increased plasma hydroxyproline in response to GH excess has not been previously reported, so this finding is novel but expected.

Glycine metabolism is a complex process, as glycine can be synthesized from multiple molecules and can be converted into numerous others. As mentioned above, a major source of circulating glycine is the diet [22]. Thus, the diet is unlikely to have a major effect on the results of this experiment, as all of the mice were given ad libitum access to the same diet. bGH mice, which show the lowest circulating levels of glycine, consume the largest amount of food [8] and have the larger intestinal surface area for absorption [26], both of which point to higher levels of circulating glycine if dietary glycine levels are the driving force.

Another possible explanation for increased plasma glycine in GHRKO mice is a decrease in glycine catabolism or excretion. Glycine can be converted to glutathione through GSS or cleaved in a process involving GLDC [22]. The expression of GSS in the liver is unchanged, while the only significant change in liver GLDC expression is a decrease in bGH males, which would indicate a decrease in glycine cleavage in bGH mice that does not correspond to the decreased plasma glycine seen in those animals. In terms of glycine excretion, there was no change in fecal glycine levels in either bGH or GHRKO mice, eliminating fecal excretion as a possible explanation for the changes in plasma glycine.

GH action is not the only thing that differentiates bGH and GHRKO mice, as they have substantial differences in body size, body composition, IGF-1 and glucose metabolism, and kidney and liver function [27,28]. bGH mice have increased body weight, decreased fat mass, and increased systolic blood pressure [29]. These mice are also hyperinsulinemic at young ages but become hypoglycemic and hypoinsulinemic at older ages [30] while having impaired kidney function [31,32] and increased ALT [33], indicative of liver damage. GHRKO mice have decreased body weight and are obese [8] but are extremely insulin sensitive [34]. They have normal kidney morphology [35] and ALT levels [36].

These phenotypic differences could also drive the glycine alterations instead of a GH-specific difference. To address this possibility, we assessed the response of glycine metabolism to acute GH treatment. When GH deficient mice (GH knockout) are treated with human GH, there is a significant decrease in both Shmt1 and Shmt2 gene expression in the liver, resulting in decreased serum glycine, demonstrating a direct response of glycine metabolism to GH treatment. Human GH is less potent in mice than bGH [37,38] but also binds to the prolactin receptor, so it is not a perfect comparison to bGH, and it is possible that the lactogenic effects of hGH may drive the gene expression changes caused by GH injection.

This acute response to GH, in conjunction with the decreased expression of SHMT1 and SHMT2 in bGH females and males, respectively, and the increased expression of SHMT2 in GHRKO males, suggests that GH action alters glycine levels in part by modulating glycine synthesis directly or through its downstream effects, including IGF-1 expression. IGF-1 changes occur in tandem with the GH alterations used for this study, so the possibility of IGF-1 (instead of GH) being the driving force for the resulting glycine and hydroxyproline changes cannot be excluded.

Increased glycine intake has been shown to extend lifespan in both male and female mice [13], possibly due to an anti-inflammatory effect [39], improved methionine clearance, or a reduction in certain types of cancer [13]. Furthermore, obesity-related diseases, such as type 2 diabetes and non-alcoholic fatty liver disease (NAFLD), are associated with lower circulating glycine levels in humans [22,40]. In fact, it has been shown that patients with morbid obesity have reduced glycine synthesis via SHMTs as well as increased insulin levels [41]. Moreover, serum glycine levels, glycine synthesis, body weight, and insulin resistance improve after bariatric surgery [41]. In addition, hydroxyproline and collagen deposition has been positively associated with the pathogenesis of several diseases. To better understand if there is an association between the lifespan seen in GHRKO and bGH mice with the plasma levels of glycine and hydroxyproline, a literature search to find papers that reported the median lifespan of male and female GHRKO and bGH mice with their respective controls was performed. We found a positive association between lifespan and plasma glycine in both bGH and GHRKO mice and a negative association between lifespan and plasma hydroxyproline in both models (Figure 4) [9,11]. These results suggest that the changes in lifespan associated with GH alterations may be associated with shifts in circulating amino acids seen in these mice.

In conclusion, excess GH increases while GH resistance decreases plasma glycine levels, and these alterations may be due to the modulation of glycine synthesis and not the catabolic pathway as shown in the SHMTs gene expression in bGH, GHRKO, and GH-injected GHKO mice. Nevertheless, glycine metabolism is quite complex, and many other proteins (not assessed here) are involved in glycine metabolism and may be altered in response to GH action. Thus, additional research is required to elucidate the process by which plasma glycine levels are lowered as a result of GH action. In addition, the consistent changes to hydroxyproline in response to GH suggest that hydroxyproline may be useful as a biomarker for GH action. The increased plasma glycine in GHRKO mice, taken together with previously published data on their longevity, agrees with previous reports showing that glycine supplementation increases lifespan. Taken together, these results indicate that there may be a shared mechanism between the two methods of extending lifespan.

## Figures and Tables

**Figure 1 metabolites-13-00191-f001:**
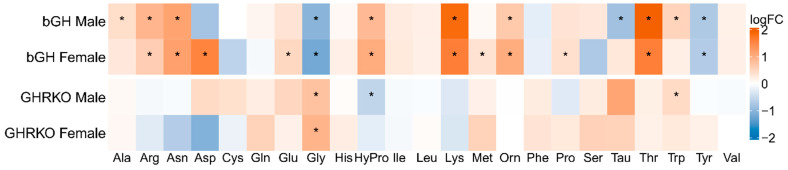
Plasma Amino Acid Levels. The plasma from 7-month-old bGH and GHRKO males and controls were used for amino acid measurements. Data are shown as the log2 fold change in each group compared to its respective control group, with red indicating an increased plasma concentration and blue indicating a decrease compared to wild type. * Indicates *t*-test *p* < 0.05 comparing each experimental group to sex-matched WT.

**Figure 2 metabolites-13-00191-f002:**
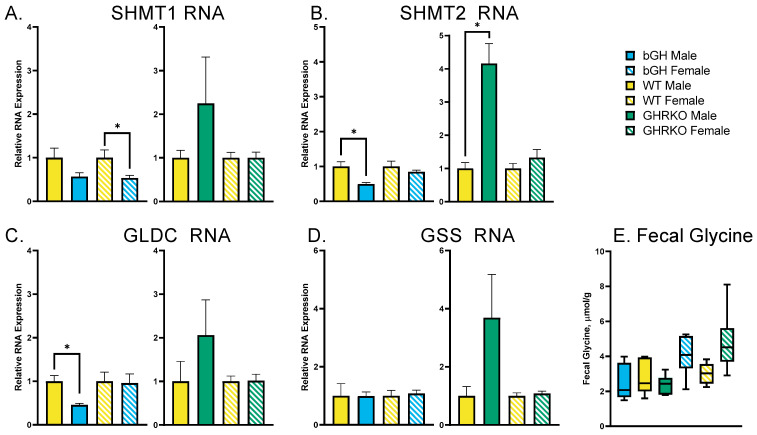
Glycine synthesis and disposal. (**A**–**D**) Liver RNA expression of genes associated with glycine synthesis (**A**,**B**) or catabolism (**C**,**D**) in male and female bGH mice and littermate WT controls and GHRKO mice and littermate WT controls. (**E**) Fecal concentration of glycine in male and female bGH and GHRKO mice and controls. * indicates *t*-test *p* < 0.05 comparing each experimental group to sex-matched littermate WT.

**Figure 3 metabolites-13-00191-f003:**
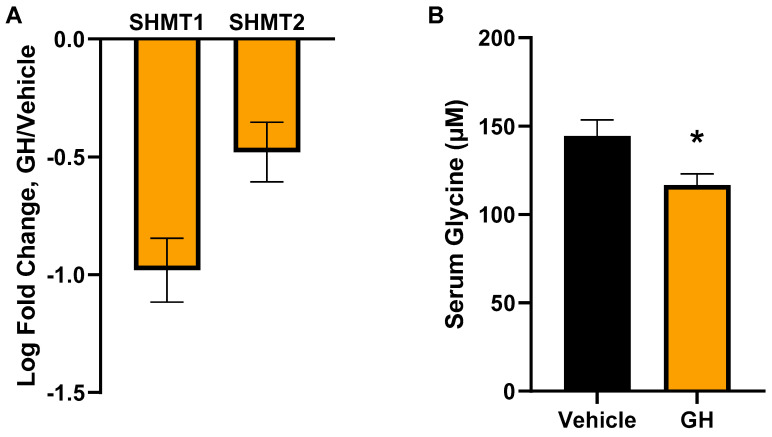
Acute effects of GH injection on glycine metabolism. (**A**) Liver RNA expression of glycine synthesis genes in male GH deficient (GHKO) mice injected with human GH or vehicle daily for five days. Data are expressed as the log2 fold change of GH-injected mice compared to vehicle-injected mice. (**B**) Serum glycine levels of the GH or vehicle-injected mice measured with a fluorometric assay. * Indicates *t*-test *p* < 0.05.

**Figure 4 metabolites-13-00191-f004:**
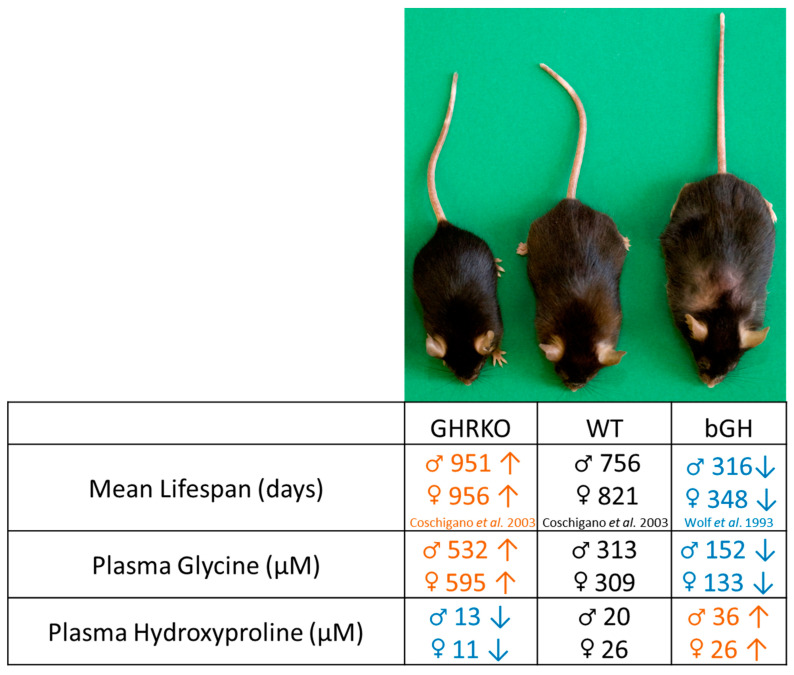
Association between plasma amino acid levels and lifespan. This image depicts the two primary mouse models examined in this study. The table shows the average lifespan of each mouse model, as reported previously [9,11], compared to the plasma levels of amino acids reported in this study. Arrows indicate *p* < 0.05 compared to WT. The WT lifespan is from the control group of the GHRKO study, as that study utilized the same genetic background (C57Bl/6J) as this experiment.

## Data Availability

The data are available from the author upon request. Data is not publicly available due to privacy or ethical restrictions.

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
