# Peer review of "Growth Hormone Alters Circulating Levels of Glycine and Hydroxyproline in Mice"

_metabolites, 2023, doi:10.3390/metabo13020191_

Round 1
Reviewer 1 Report
Growth hormone has established effects on protein metabolism, such as increasing protein synthesis and decreasing amino acid degradation. Lifespan extension in mice has also been accomplished through the modulation of dietary amino acids. However, the effects of growth hormone on amino acid metabolism are not fully understood. Thus, the aim of this study was to determine whether growth hormone action altered circulating amino acid levels in mice models. When growth hormone action was increased, as in bovine growth hormone transgenic mice, plasma glycine levels were decreased. In contrast, plasma glycine increased in growth hormone receptor knockout mice. Acute growth hormone treatment of growth hormone knockout mice caused decrease of Shmt1 and Shmt2 expression in liver, as well as decrease in serum glycine. The authors conclude that excess growth hormone decreases plasma glycine levels and these alterations may be due to modulation of glycine synthesis and not the catabolic pathway. The manuscript is well-written and the methods sound. I did not have any major concerns, only minor issues listed below:
Page 1 lines 13, “Growth Hormone (GH) has established effects on protein metabolism, such as increasing protein synthesis and decreasing amino acid degradation, but its effects on circulating amino acid levels are less studied.”, Growth hormone.
Page 1 lines 20, “bGH mice have significantly decreased plasma glycine and increased hydroxyproline in both sexes, while GHRKO mice have increased plasma glycine in both sexes and decreased hydroxyproline in males.”, The past tense is the natural choice when describing the results obtained.
Page 1 lines 22, “Glycine synthesis gene expression is decreased in bGH mice (Shmt1 in males and Shmt2 in females) and increased in GHRKO mice (Shmt2 in males).”, “is”
Page 1 lines 24, “Acute GH treatment of GHKO mice causes decreased liver Shmt1 and Shmt2 expression and decreased serum glycine.”, Please revise this sentence.
Page 7 lines 280, “Furthermore, obesity related diseases.”
Author Response
Thank you for your feedback. We have made the following changes:
Page 1 lines 13, “Growth Hormone (GH) has established effects on protein metabolism, such as increasing protein synthesis and decreasing amino acid degradation, but its effects on circulating amino acid levels are less studied.”, Growth hormone.
We have made the requested change.
Page 1 lines 20, “bGH mice have significantly decreased plasma glycine and increased hydroxyproline in both sexes, while GHRKO mice have increased plasma glycine in both sexes and decreased hydroxyproline in males.”, The past tense is the natural choice when describing the results obtained.
We have made the requested change.
Page 1 lines 22, “Glycine synthesis gene expression is decreased in bGH mice (Shmt1 in males and Shmt2 in females) and increased in GHRKO mice (Shmt2 in males).”, “is”
We have made the requested change.
Page 1 lines 24, “Acute GH treatment of GHKO mice causes decreased liver Shmt1 and Shmt2 expression and decreased serum glycine.”, Please revise this sentence.
We have made the requested change.
Page 7 lines 280, “Furthermore, obesity related diseases.”
We have removed the superfluous period.
Reviewer 2 Report
ID: metabolites-212202
Growth Hormone Alters Circulating Levels of Glycine and Hydroxyproline. by Jonathan et al.
To the Authors:
GENERAL COMMENTS:
The authors investigated the effects of GH on circulating levels of amino acids in mice. They first showed that GH action was negatively associated with plasma glycine levels and positively associated with plasma hydroxyproline levels in bGH and GHRKO mice. It was demonstrated that glycine synthesis gene expression was decreased in bGH mice (Shmt1 in females and Shmt2 in males) and increased in GHRKO mice (Shmt2 in males). It was also revealed that five daily injections of human GH in GHKO mice decreased expression of glycine synthesis genes (Shmt1 and Shmt2) and serum glycine levels.
This manuscript contains some interesting points regarding the relationships of GH and circulating amino acids (glycine and hydroxyproline) in mice, but several issues need to be confirmed to conclude the present findings.
SPECIFIC COMMENTS:
1. IGF-I, a major mediator of GH, is not mentioned and examined at all. The functional involvement of IGF-I in the effects of GH on circulating glycine and hydroxyproline needs to be investigated particularly, in the liver.
2. This study suggests that the effects of GH on glycine synthesis gene expression partly in a GH receptor-mediated manner, but what about changes of the GH signaling?
3. The results are solely derived from the knockout mice. The title should indicate that the conclusions were based on the experimental mice data.
4. The data regarding the metabolic states and liver functions in these experimental mice should be provided. For example, the changes of body weight, BMI, blood pressure, liver and renal functions (AST, ALT, LDH, UN, CRTN, etc.), and glucose, insulin and lipids levels should be shown in the results.
5. Correct that the results of glycine synthase gene expression in bGH mice are reversed for males and females in the abstract on line 23.
Author Response
SPECIFIC COMMENTS:
- IGF-I, a major mediator of GH, is not mentioned and examined at all. The functional involvement of IGF-I in the effects of GH on circulating glycine and hydroxyproline needs to be investigated particularly, in the liver.
Thank you for the feedback. Although we mentioned IGF-1 in the introduction, we agree that the discussion lacked consideration of IGF-1’s role in these experiments. We have added information to the discussion (lines 283-286) pointing out that IGF-1 could also be the driving force of the results of this study, as IGF-1 changes were not disentangled from GH changes in this study.
- This study suggests that the effects of GH on glycine synthesis gene expression partly in a GH receptor-mediated manner, but what about changes of the GH signaling?
Unfortunately, we are not sure what the reviewer means with this question. If they are asking about the intracellular signaling of the GH receptor (JAK2 or STAT5b phosphorylation, etc.), we have not measured this in the mice used for this study, and unfortunately, the mice are no longer available for further experimentation.
- The results are solely derived from the knockout mice. The title should indicate that the conclusions were based on the experimental mice data.
We have added “in Mice” to the title to make it clear which species the results were obtained from.
- The data regarding the metabolic states and liver functions in these experimental mice should be provided. For example, the changes of body weight, BMI, blood pressure, liver and renal functions (AST, ALT, LDH, UN, CRTN, etc.), and glucose, insulin and lipids levels should be shown in the results.
We agree that those data would be interesting to include; however, these mice are not available for further measurements. Fortunately, the growth and metabolic states of bGH and GHRKO mice have been extensively reported elsewhere and we have added this information to the discussion in lines 267-272.
- Correct that the results of glycine synthase gene expression in bGH mice are reversed for males and females in the abstract on line 23.
Thank you for bringing this error to our attention. We have corrected it.
Reviewer 3 Report
The main question that is addressed by the authors is the effect of bovine growth hormone on circulating levels of amino acids such as glycine and hydroxyproline in mice and GHR knockout mice. The topic is considered original as it investigates significant markers that are critical for disease states such as acromegaly. As the manuscript is designed based on acute drug administration it sheds a light on the side effects. If the authors would be able to compare human GH and bovine GH it improves the quality of the manuscript. In this way they could use the advantage of some docking studies to define the affinity towards the target or retrieve the data from other studies to enrich their references. However, the conclusions are consistent with the evidence and arguments presented and they address the main question posed. The manuscript can be reconsidered after major revision following the response to the comments
1- Why bovine GH and not human GH is injected
2- Please name the GH types and brand names (with dose and duration of injection) that are available in the pharmaceutical market currently
3- Please address the affinity of bovine GH, mice GH, and human GH towards the mice GHR (receptor) and define how it might affect the potential observation
Author Response
1. Why bovine GH and not human GH is injected
Thank you for this question. The GH injection was carried out as component of another study that required the use of human GH rather than bovine or mouse GH. While bGH would have been ideal for comparison purposes, we had limited access to that resource. A major difference between bovine GH and human GH is that human GH also binds to the prolactin receptor, while bGH binds only to GHR. Because bGH transgenic mice had decreased serum glycine, we know that the glycine alterations stimulated by GH do not require the prolactin receptor-binding activity of GH. We have added to the discussion (lines 282-284) the possibility of prolactin receptor binding by human GH.
2- Please name the GH types and brand names (with dose and duration of injection) that are available in the pharmaceutical market currently
This experiment used human GH from Protein Laboratories Rehovot (Rehovot, Israel). The GH from this company is produced in Escherichia coli and purified, as previously described (Solomon et al 2006). The GH was reconstituted in 0.4% NaHCO3 and mixed by gentle inversion to a concentration of 500 ug/ml and was administered by subcutaneous injection. This dose of GH was based on previous dosing studies in C57BL/6J mice required for observable changes to body composition and IGF-1 (List et al 2009). We have added this information to the methods section (line 132). When prescribed by a licensed physician, the pharmaceutical choices for GHD humans are somatropin marketed by Genentech (Nutropin), Eli Lilly (Humatrope), Pfizer (Genotropin), Novo Nordisk (Norditropin), Teva (Tev-Tropin), and Merck (Saizen). Additionally, two long-acting GH analogs have been FDA approved: somapacitan-beco, and lonapegsomatropin-tcgd.
3- Please address the affinity of bovine GH, mice GH, and human GH towards the mice GHR (receptor) and define how it might affect the potential observation
Bovine GH has a higher affinity for the mouse GH receptor (Chen et al 1991) and is 2.1 times more potent in mice than human GH (Nicoll et al 1986). Comparisons between mouse GH and human or bovine GH are lacking. We have added this to the discussion in lines 282-284.
Round 2
Reviewer 2 Report
To the Authors:
GENERAL COMMENTS:
Thank you for your revision; however, the lack of experiments regarding IGF-I leads to the difficulty in determining the precise effects of GH on protein metabolism.
Author Response
Thank you for your revision; however, the lack of experiments regarding IGF-I leads to the difficulty in determining the precise effects of GH on protein metabolism.
We agree that our experiments do not distinguish between GH’s direct and IGF-mediated or indirect effects. This problem is common to the many studies that explore the effects of GH in these mice. We mentioned this shortcoming in the discussion previously, and we have now included additional information in the introduction to reinforce that limitation (line 65). Further, we acknowledge that future studies should work to address this; however, those experiments are outside the scope of this initial study and are not feasible in our laboratory.
Reviewer 3 Report
The comments are addressed properly.
Round 3
Reviewer 2 Report
To the Authors:
GENERAL COMMENTS:
Thank you for your revision. Regarding my concerns on this study, most of the part were figured out.